# Enhanced Recovery after Uterine Corpus Cancer Surgery: A 10 Year Retrospective Cohort Study of Robotic Surgery in an NHS Cancer Centre

**DOI:** 10.3390/cancers14215463

**Published:** 2022-11-07

**Authors:** Christina Uwins, Radwa Hablase, Hasanthi Assalaarachchi, Anil Tailor, Alexandra Stewart, Jayanta Chatterjee, Patricia Ellis, Simon S. Skene, Agnieszka Michael, Simon Butler-Manuel

**Affiliations:** 1Academic Department of Gynaecological Oncology, Royal Surrey NHS Foundation Trust, Guildford GU2 7XX, UK; 2Swansea Gynaecological Oncology Centre (SGOC), Swansea Bay University Health board, Singleton Hospital, Swansea SA2 8QA, UK; 3St. Luke’s Cancer Centre, Royal Surrey NHS Foundation Trust, Guildford GU2 7XX, UK; 4School of Biosciences and Medicine, University of Surrey, Guildford GU2 7XH, UK; 5Surrey Clinical Trials Unit, University of Surrey, Guildford GU2 7XP, UK

**Keywords:** endometrial cancer, robotic surgery, Da Vinci, ERAS, uterine cancer, minimally invasive surgery

## Abstract

**Simple Summary:**

Surgical and survival outcomes for uterine corpus cancer following the introduction of robotic surgery to Royal Surrey NHS Foundation Trust; a large volume United Kingdom teaching hospital and cancer centre. Introduction of the Da Vinci^TM^ robot was associated with enhanced recovery after surgery with low 30-day mortality (0.1%), low return to theatre (0.5%), a low use of blood transfusion and intensive care (1.8% & 7.2% respectively), low conversion to open surgery (0.5%) and a reduction in median length of stay, with comparable survival to published data, and a three to four fold increase in cases treated. This increased productivity was associated with a highly predicable patient pathway of care, for high-risk patients, with reduced demands on health services.

**Abstract:**

Royal Surrey NHS Foundation Trust introduced robotic surgery for uterine corpus cancer in 2010 to support increased access to minimally invasive surgery, a central element of an enhanced recovery after surgery (ERAS) pathway. More than 1750 gynaecological oncology robotic procedures have now been performed at Royal Surrey NHS Foundation Trust. A retrospective cohort study was performed of patients undergoing surgery for uterine corpus cancer between the 1 January 2010 and the 31 December 2019 to evaluate its success. Data was extracted from the dedicated gynaecological oncology database and a detailed notes review performed. During this time; 952 patients received primary surgery for uterine corpus cancer; robotic: *n* = 734; open: *n* = 164; other minimally invasive surgery: *n* = 54. The introduction of the Da Vinci^TM^ robot to Royal Surrey NHS Foundation Trust was associated with an increase in the minimally invasive surgery rate. Prior to the introduction of robotic surgery in 2008 the minimally invasive surgery (MIS) rate was 33% for women with uterine corpus cancer undergoing full surgical staging. In 2019, 10 years after the start of the robotic surgery program 91.3% of women with uterine corpus cancer received robotic surgery. Overall the MIS rate increased from 33% in 2008 to 92.9% in 2019. Robotic surgery is associated with a low 30-day mortality (0.1%), low return to theatre (0.5%), a low use of blood transfusion and intensive care (1.8% & 7.2% respectively), low conversion to open surgery (0.5%) and a reduction in median length of stay from 6 days (in 2008) to 1 day, regardless of age/BMI. Robotic survival is consistent with published data. Introduction of the robotic program for the treatment of uterine cancer increased productivity and was associated with a highly predicable patient pathway of care, for high-risk patients, with reduced demands on health services. Future health care commissioning should further expand access to robotic surgery nationally for women with uterine corpus cancer.

## 1. Introduction

The incidence of uterine corpus cancer (corpus cancer) is increasing; partly due to rising obesity and the increasing incidence of type 2 diabetes [1]. Increased body mass index (BMI) increases surgical complexity and is associated with increased peri-operative complications including increased intra-operative blood loss, anaesthetic complications, infection, venous thromboembolism and wound dehiscence in open surgery [2,3,4]. Minimally invasive surgery (MIS) is the recommended route of surgery for endometrial cancer as it mitigates these risks [5,6].

Minimally invasive surgery is one of the key principals of enhanced recovery after surgery (ERAS) [7]. Data from two studies has shown that for women undergoing minimally invasive surgery for endometrial cancer, the risks are similar for those with a BMI greater than 40 to those with a BMI less than 40 [8,9]. Conversion rates to open can be high in this population, particularly when performing full surgical staging [10,11,12,13,14]. Robotic surgery in morbidly obese patients has been associated with better quality staging with minimal conversions to laparotomy compared to laparoscopy [14,15].

Compared to laparotomy robotic staging is associated with less post-operative morbidity with no evidence of difference in survival outcomes in retrospective studies [16,17]. In an experienced laparoscopic centre; Mäenpää et al. (2016) completed a randomised controlled trial comparing robotic surgery (*n* = 50) to laparoscopy (*n* = 49) [18]. Shorter operative time and lower conversion rates to open surgery were observed in patients undergoing robotic surgery compared to laparoscopic surgery. A systematic review and meta-analysis of robotic surgery for endometrial cancer by Wang et al. (2020) found that robotic surgery was associated with less blood loss and blood transfusion, shorter length of stay, fewer conversions and complications and no difference in lymph node yield [19]. The mechanical and ergonomic support provided by robotic surgery aids reduced conversion rates to open surgery [19,20].

Overall non-standardised 5-year survival of women diagnosed with endometrial cancer in England is 92.5% for stage I disease to 14.9% with stage IV disease at presentation. For women 75 years and older this drops to 87.9% in stage I disease to 8.8% for stage IV disease [21]. The elderly are often surgically under-staged despite often presenting with higher-grade tumours [22,23]. There is no evidence of an increased incidence of perioperative complications due to age alone. All women should have equal access to full surgical staging and the benefits of MIS [24,25,26,27,28,29].

Royal Surrey NHS Foundation Trust introduced robotic surgery in 2009. Since then the academic department of gynaecological oncology has performed more than 1750 robotic procedures. Robotic surgery for uterine corpus cancer commenced in 2010 to support increased access to minimally invasive surgery, a central element of an enhanced recovery after surgery (ERAS) pathway. The aim of this study was to assess the surgical and survival outcomes for uterine corpus cancer surgery since robotic surgery was implemented and evaluate its success. This paper presents the experience of implementing robotic surgery for uterine corpus cancer in a high volume UK cancer centre with detailed breakdown of both stage and grade of disease and “real world” survival statistics.

## 2. Materials and Methods

Retrospective cohort study of sequential surgical treatment for primary corpus cancer (defined as having had a hysterectomy performed for corpus cancer) performed at Royal Surrey NHS Foundation Trust between 1 January 2010 and 31 December 2019. Data was obtained from the dedicated gynaecological oncology departmental database where operation notes and any complications are recorded contemporaneously. These data were cross-referenced with blood transfusion and intensive care databases to provide data on the use of blood transfusion and intensive care facilities. Data from 2008 and 2009 was also examined to provide a historical baseline of practice prior to the introduction of the Da Vinci^TM^ robot. Diagnostic procedures and surgery for recurrent disease were excluded. Missing data were sought through review of the notes where available. Whole departmental data is presented principally representing the work of 4 consultants and their senior trainees. Operative technique and use of uterine manipulators was not standardised between operating surgeons. In 2010 robotic surgery was performed using the Da Vinci ^TM^ S system (Intuitive Surgical, Sunnyvale, CA, USA). In 2015 the Da Vinci^TM^ S were replaced with 2 Da Vinci^TM^ Si with the Intuitive Firefly fluorescence imaging system.

A cohort of 952 women who had received primary surgery between the 1 January 2010 and the 31 December 2019 for corpus cancer was identified and subdivided into three separate cohorts. The robotic cohort included all patients who underwent surgery using a Da Vinci^TM^ robot and includes all cases of conversion to open surgery after docking of the robot. The open surgery cohort was defined as any patient undergoing a laparotomy with the exclusion of those whose surgery started by the robotic, laparoscopic or vaginal surgical route. The Other MIS group was formed of those not in the robotic or open cohort and includes total laparoscopic hysterectomy (48), laparoscopic assisted (3) and vaginal hysterectomies (3). Core outcomes sought included: age, American Society of Anesthesiologists (ASA) physical status classification system, Body Mass Index (BMI), tumour type, Stage and grade, lymph node sampling/dissection location, Omentectomy/biopsy, estimated blood loss (EBL), length of stay (LOS), return to theatre <30 days, 30 day mortality, use of blood transfusion, use of intensive care unit (ITU), 30 day morbidity by Clavien Dindo classification, conversion to open for the MIS and robotic cohort and length of surgery. To investigate the effect of BMI or age on surgical outcomes the robotic cohort was subdivided by BMI (7 groups: <25, 25–29.9,30–34.9,35–39.9, 40–44.9, 45–49.9, 50+) and age (7 groups: <55, 55–59, 60–64, 65–69, 70–74, 75–79, 80+). Data on length of surgery was calculated from operation start time (surgeons scrubbed and starting to prep patient) or from completion of vaginal phase (occurring just before laparoscopic phase in robotic assisted procedures). Data extracted from the gynaecological oncology database was analysed in Microsoft Excel to provide descriptive statistics. Cause of death was determined as due to disease or not following review of the medical records and contacting general practitioners or the coroner where appropriate. A minimum of 2 years follow up is presented. To investigate the oncological impact of the robotic program, survival analysis of our robotic cohort was carried out. Additionally to enable comparison with published studies a subset analysis of survival and recurrence free survival for women with stage 1 endometrioid endometrial carcinoma (all grades) was performed. Survival curves were generated with the use of the Kaplan–Meier method based on date of diagnosis.

## 3. Results

Between 1 January 2010–31 December 2019, 952 patients underwent primary surgery for corpus cancer. Table 1 describes the patient demographics, surgical procedures and outcomes. Using the Da Vinci^TM^ robotic system; 734 procedures were performed, with an overall conversion rate of 0.5%, after docking robot, or 1.7% if conversions to laparotomy following initial laparoscopic assessment are included (robot not docked). Median estimated blood loss (EBL) for the robotic cohort was 50mL, median Length of stay (LOS), 1 day and 30-day mortality was 1/734 (0.1%). Open surgery was performed in 164 cases with a median EBL of 500mL, median length of stay 6 days, 30-day mortality 5/164 (3.0%). In this 10-year period 54 patients underwent total laparoscopic hysterectomy (48), laparoscopic assisted (3) or vaginal hysterectomy (3) This “Other MIS” group had a median EBL of 100 mL, median length of stay 2 days, 30 day mortality 1/54 (1.9%). Figure 1 shows the relationship between the numbers of cases of robotic surgery for corpus cancer to length of stay and blood loss.

In 2008 the median length of stay for women with corpus cancer was 6 days, median EBL 300 mL with 33.3% (11/33) of operations for corpus cancer performed laparoscopically. In 2019 91.3% (115/126) of all operations performed for corpus cancer were performed using the Da Vinci^TM^ robot with 9 performed open (predominately due to bulky uterine size) and 2 laparoscopically. Between 2008–2019 the overall median length of stay for all patients with corpus cancer fell, from 6 days to 1 night (Figure 1). The total rate performed by MIS increased from 33.3% to 92.9% and the overall conversion rate from minimally invasive surgery to open fell from 18% in 2008 to 1.7% in 2019 despite increasing numbers of patients with obesity. Annual numbers of procedures performed for corpus cancer increased from 33 to 126 in this time period.

### 3.1. Body Mass Index (BMI)

Median BMI in our robotic cohort was 31.1 (range 16.4–75.2). The median EBL in our robotic cohort increases only minimally with increasing BMI with a median EBL in the BMI 50+ cohort of 150 mL (range 20–1900). Median length of stay is constant at 1 day across all BMI groups (Figure 2). Three out of the four conversions to laparotomy in the robotic cohort occurred in the BMI < 40 groups (Table 2).

### 3.2. Age

Within the robotic cohort at Royal Surrey, median length of stay remains constant at 1 day as does median blood loss of 50 mL across all age groups (Table 3).

### 3.3. Length of Surgery

Data on length of surgery was available for 93.1% of patients. The median length of surgery for the robotic cohort was shorter in the final five years at 2 h 37 min compared to 2 h 54 min over the whole 10-year period studied. Median length of surgery for open was 2 h 50 min and other MIS 2 h 54 min (Table 4).

### 3.4. Survival

Five-year overall and relapse-free survival, stratified by stage, for the robotic cohort is represented in Figure 3. Overall survival and relapse free survival at 4.5 years for stage I and stage I and II combined endometrioid endometrial carcinoma in our robotic cohort was additionally calculated to enable benchmarking against published studies. Overall survival for stage 1 endometrioid endometrial carcinoma at 4.5 years was 90.3% (CI = 86.8–93.8% (*n* = 481) and 90.2% (CI = 86.8–93.6%) (*n* = 519) for stage I and II endometrioid endometrial carcinoma combined. Relapse free survival at 4.5 years for stage I endometrioid endometrial carcinoma in our robotic cohort is 86.4% (CI = 82.5–90.4) (*n* = 481) and 86.6% (CI = 82.9–90.4%) (*n* = 519) for endometrioid endometrial carcinoma stage I and II combined.

## 4. Discussion

The use of robotic surgery has steadily grown since its introduction to Royal Surrey NHS Foundation Trust (Figure 4). Initially access to the robotic system was limited with access improving over time. More complex/high risk uterine corpus cancer surgeries are performed at Royal Surrey NHS Foundation Trust, the cancer centre, with surgery for early stage low-grade disease performed at local unit hospitals. A higher proportion of women with raised BMI, complex comorbidities or high-risk cell types therefore receive surgery at Royal Surrey NHS Foundation Trust than would otherwise be seen at a population level. This is in line with NHS implementing outcomes guidance and has resulted in our annual numbers of procedures for corpus cancer increasing more than 3-fold since 2008 (Table 1 & Figure 1).

Endometrioid endometrial cancer was the causative cell type in 77.4% of women undergoing robotic surgery for their corpus cancer. This figure closely matches that published by NCIN in 2013 (76.9%) as representative of the national incidence of the endometrioid tumour type at the time (Table 1) [30]. The open cohort had a greater proportion of high stage/grade disease than both the robotic and other MIS cohorts. This is potentially a result of high-grade disease presenting at an advanced stage being referred for full surgical staging and cytoreductive surgery.

The Laparoscopic approach to cancer of the Endometrium (LACE) trial was a multinational randomized equivalence trial conducted between October 2005 and June 2010. Women with stage I endometrioid endometrial cancer (any grade) were randomised 1:1 to either total laparoscopic or total abdominal hysterectomy [10]. The Denmark population data published by Jørgensen et al. (2019) assessed the survival estimates of women with stage I-II endometrial cancer who underwent surgery between 1 January 2005–30 June 2015 [31]. Direct comparison is imperfect, partly due to not being able to compare small subgroups with available population survival data but these internationally recognised studies do provide us with a benchmark for us to compare our local outcomes.

Comparing our data to that of the LACE trial; the stage I endometrioid endometrial adenocarcinoma robotic group had fewer grade 1 and more grade 3 tumours than seen in LACE overall (Table 1). The LACE trial reported a 4.5 year overall survival rate of 92.4% (laparotomy *n* = 353) vs. 92.0% (laparoscopy *n* = 407). Overall survival at 4.5 years for stage I endometrioid endometrial carcinoma in our robotic cohort is comparable at 90.3% (CI = 86.8–93.8% (*n* = 481) and 90.2% (CI = 86.8–93.6%) (*n* = 519) for endometrioid endometrial carcinoma stage I and II combined. With regard to relapse free survival LACE reported relapse-free survival at 4.5 years of 81.3% in the total abdominal hysterectomy group and 81.6% in the total laparoscopic hysterectomy group. Relapse free survival at 4.5 years for stage I endometrioid endometrial carcinoma in our robotic cohort is 86.4% (CI = 82.5–90.4%) (*n* = 481) and 86.6% (CI = 82.9–90.4%) (*n* = 519) for endometrioid endometrial carcinoma stage I and II combined.

Royal Surrey NHS Foundation Trust was an early adopter and pioneer of enhanced recovery after surgery [32,33,34,35]. The program and associated pathways were established by the time the first robotic system was installed. Over the 10 years this study covers, the median length of stay for robotic surgery reduced from 2 days in 2010 to 1 day in 2011, the second year after its introduction. Median length of stay for women undergoing robotic surgery for uterine cancer has remained consistent year by year, at 1 day, ever since regardless of age or BMI. Since 2008 length of stay for the other MIS group fluctuated between 1 and 4 days with an overall median length of stay of 2 days over the 10 years of the study. This is a reflection of the relatively small numbers of traditional laparoscopic procedures per year and the influence that laparoscopic conversions to open surgery, then has on median length of stay. Since 2008 length of stay for open surgery reduced from 7 to 5 days. The moderate reduction in length of stay in the other MIS and open surgery groups may reflect the maturing/development of the ERAS pathway over this time period, The retrospective nature of this study can, however, only illuminate possible associations rather determine causation.

A detailed health economic assessment directly relating to the 10 years studied is beyond the scope of this study. The economic cost following the introduction of the robotic program has, however, been analysed at a trust level. The potential cost saving through cost modelling using the 2019–2020 national HES data and the departmental open, laparoscopic and robotic data (including assessment of length of stay, complications, readmission rate and rate of conversion to open surgery) has demonstrated a cost saving of £2442 when robotic surgery is compared to open and £651 when compared to laparoscopy. The reduction in post-operative 30 day complication rate by performing robotic procedures on these patients has also demonstrated a cost saving of £763 vs. open and £161 vs. laparoscopic procedures. Due to the reduction in conversion rates cost modelling revealed a potential saving for £671 when comparing robotic vs. laparoscopic procedures [36,37,38,39,40,41,42]. Following the introduction of extended user pricing by Intuitive surgical ltd since 2020 the cost of each robotic assisted procedure has reduced further by £219.

These costs are based on our standard three arm robotic procedures, performed for a woman with endometrial cancer, using cadiere forceps, bipolar and scissors, avoiding the use of a needle driver. These cost savings are a product of the high successful minimally invasive surgery rate seen with robotic surgery, reduced blood transfusions and short inpatient stay including associated reduced use of HDU and ITU. The NHS national schedule of reference costs sets the median cost of an inpatient surgical bed at £407 per night and £619 per HDU night or £1190 for an ITU bed [36,37]. Higher rates of conversion to open surgery require the use of open surgery equipment in addition to laparoscopic or robotic equipment and prolonged hospital stay. Post-operative complications are upsetting and stressful for patients and significantly impact quality of life. Complications increase heath service use, with increased investigations including imaging and blood tests and pressurise an already stretched primary healthcare system. Straatman et al. (2015) calculated the cost of a minor complication following major abdominal surgery to be EUR €6828 per patient. This is a conservative calculation as it only took into account the additional hospital costs and not those occurring outside of the hospital [40]. Loss of work or the inability to resume caring responsibilities all have their cost both personal and societal.

### Strengths and Limitations

The strength of this 10-year retrospective cohort study is that it reports the outcomes associated with the introduction of robotic surgery for corpus cancer in a large UK cancer centre with all consecutive surgical procedures for corpus cancer included. Adjuvant oncological treatments were offered to women based on the guidance at the time of their diagnosis and has changed, largely based on the findings of the ASTEC Trial and PORTEC studies, during the 10-year period that this cohort covers [43,44,45,46]. The retrospective nature of this study relies on the quality of inputted data and may be subject to information bias. Selection bias has been limited as operation notes for each procedure are produced at the time of surgery within the dedicated gynaecological oncology database ensuring that an as accurate record as possible of the route of surgery performed is recorded. Survival analysis of our robotic cohort was performed to investigate the oncological impact of the introduction of the robotic program. Whilst we acknowledge the limitations of this comparison due to the retrospective nature of this study both LACE and the Denmark population data published by Jørgensen et al. (2019) provide a well established benchmark in surgical endometrial cancer care and it is reassuring that our retrospective data appears comparable.

## 5. Conclusions

This study represents the largest single centre retrospective cohort study of robotic surgery in the United Kingdom providing a detailed breakdown of both stage and grade of disease and presenting “real world” survival statistics from a high volume UK cancer centre.

Since the introduction of robotic surgery to Royal Surrey NHS Foundation Trust, the overall minimal invasive surgery rate for uterine corpus cancer has increased from 33% in 2008 to 92.9% in 2019 and the overall conversion rate to open has reduced from 18% to 1.7% despite an increasingly obese population with allied medical co-morbidities. These results are akin to those of the Denmark population data published by Jørgensen et al. (2019) [31]. Nationally the picture has been very different with the MIS rate for endometrial cancer being 68.7% in 2017/2018 with significant variation between geographical regions [47]. Median estimated blood loss for women undergoing surgery for uterine cancer has fallen from 300 mL to 50 mL and our median overall length of stay for uterine cancer from 6 days to 1 night with comparable operating times. Median length of stay for women undergoing robotic palliative hysterectomy, for bleeding, with stage IV endometrial cancer is maintained at 1 night. Robotic surgery is particularly well suited to patients with high BMI’s; with surgical staging performed without undue difficulty or surgical compromise. For many cases previously thought not fit for surgery, robotic surgery is now offered at Royal Surrey NHS Foundation Trust.

The introduction of the Da Vinci^TM^ robot for uterine cancer surgery has led to revolutionary change in practice at Royal Surrey NHS Foundation Trust since 2010 with considerable benefit to patients in the form of enhanced recovery, and also to the hospital itself. The reliability of robotic surgery, enabling more than 90% of women to undergo minimally invasive surgery to surgically treat their uterine cancer, alongside an already established enhanced recovery program, has greatly increased hospital productivity and efficiency with predictable short length of stay. This has been achieved regardless of age or BMI, low use of blood transfusion and minimal use of costly ITU/HDU facilities despite a near four fold increase in patient numbers [32,33,35]. Our experience, which is that of a high volume UK cancer centre, has considerable beneficial implications for the NHS nationally, which is currently struggling with increasing waiting lists across all surgical services, and also for similar public health systems in other countries. The benefits of robotic surgery are already being applied and rolled out in other surgical specialties. Future health care commissioning should further expand access to robotic surgery nationally for women with uterine corpus cancer.

## Figures and Tables

**Figure 1 cancers-14-05463-f001:**
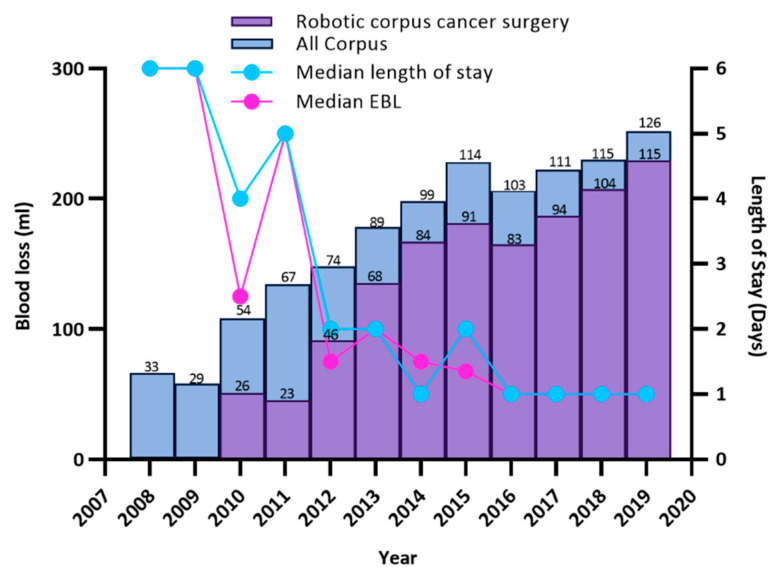
Implementation of Robotic Surgery for Corpus Cancer at Royal Surrey NHS Foundation Trust.

**Figure 2 cancers-14-05463-f002:**
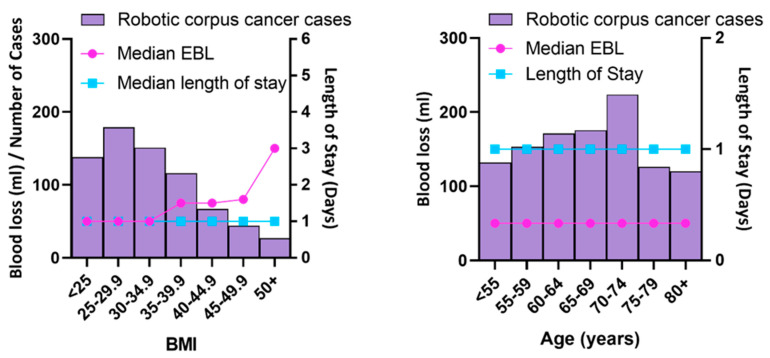
Robotic Cohort—Association between BMI/Age and Median EBL/Length of stay (LOS).

**Figure 3 cancers-14-05463-f003:**
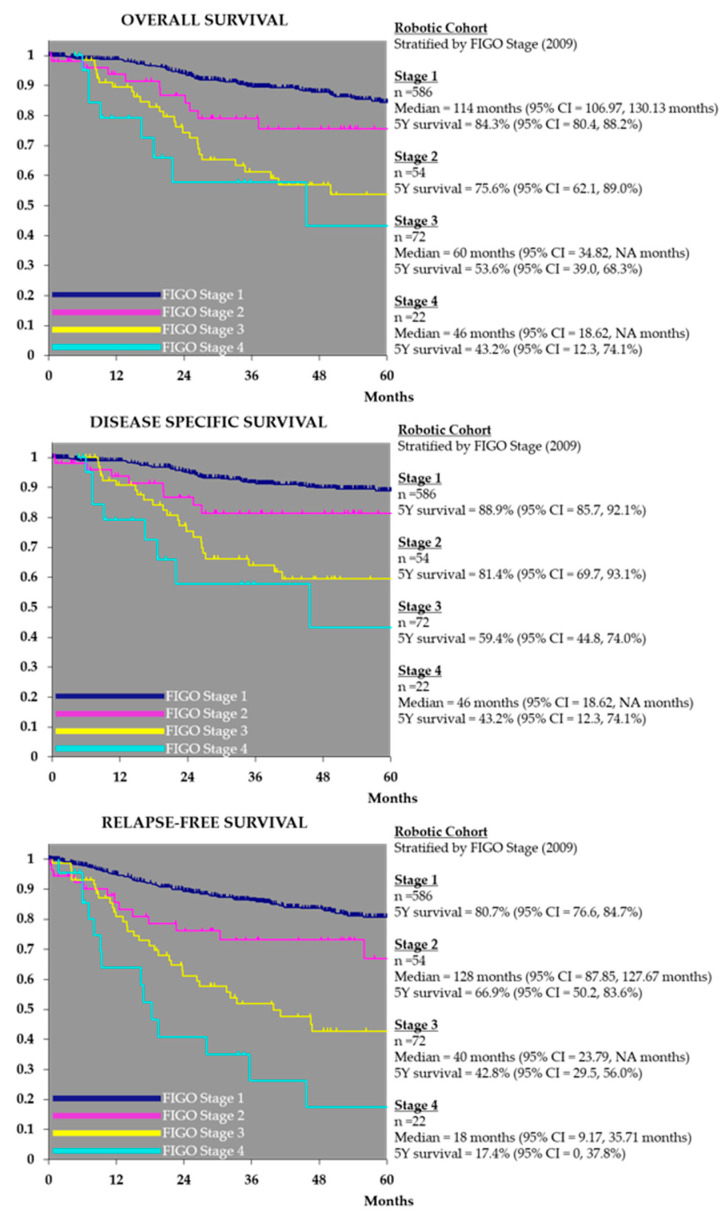
Robotic Cohort—Overall, Disease Specific Survival & Relapse Free Survival by FIGO Stage (2009).

**Figure 4 cancers-14-05463-f004:**
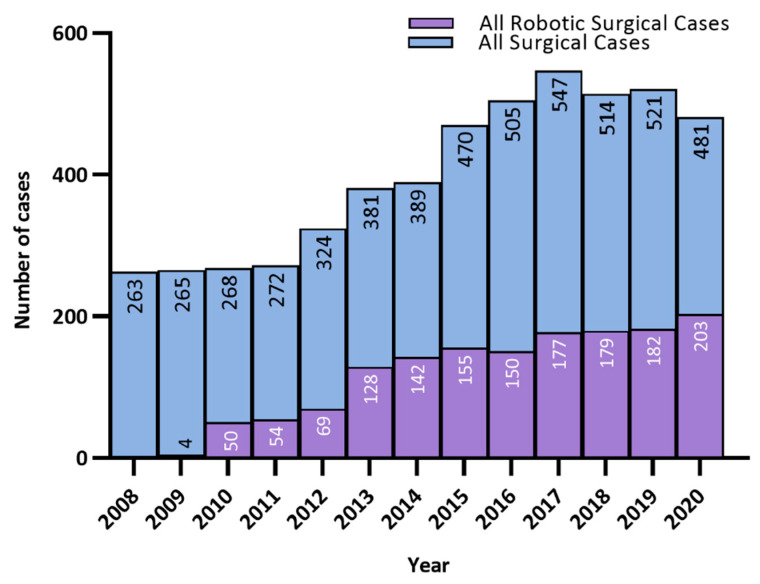
Growth in the use of Robotic Surgery at Royal Surrey NHS Foundation Trust, Guildford.

**Table 1 cancers-14-05463-t001:** Corpus Cancer Surgery 2008/2009 and 2010–2019 Patient demographics, surgical procedures and outcomes. * National Cancer Intelligence Network (NCIN) (2013). Outline of Uterine Cancer in the United Kingdom: Incidence, Mortality and Survival **^$^** LACE Trial.

Corpus Cancer Primary Urgery		2010–2019					
	2008/2009	Overall	Robotic	Other MIS	Open	Comparative Data	
Number (*n*)	62	952	734 (77.1%)	54 (5.7%)	164 (17.2%)	Laparotomy	Laparoscopy

Median Age	67.5	68.0	67.0	70.5	69.0	63.1 yrs ^$^	63.3 yrs ^$^
Age range	33–89	31–91	31–90	35–91	37–9		

Median ASA	2	2	2	2	2		
ASA range	1–4	1–4	1–4	1–3	1–4		

ASA1	35.5% (22)	9.7% (92)	10.8% (79)	5.6% (3)	6.1% (10)		
ASA2	45.2% (28)	65.0% (619)	63.8% (468)	66.7% (36)	70.1% (115)		
ASA ≥ 3	19.4% (12)	22.5% (214)	23.4% (172)	22.2% (12)	18.3% (30)		
ASA Data unavailable (*n*)	0	2.8% (27)	2.0% (15)	5.6% (3)	5.5% (9)		

BMI Median	29.9	30.5	31.1	29.0	27.7		
BMI Range	17.7–54.4	16.1–75.2	16.4–75.2	18.6–55.0	16.1–51.4		
BMI Data unavailable (*n*)	12.9% (8)	3.9% (37)	1.6% (12)	13.0% (7)	11.0% (18)		

Stage I	66.1% (41)	72.7% (692)	79.8% (586)	75.9% (41)	39.6% (65)		
Stage II	14.5% (9)	7.7% (73)	7.4% (54)	7.4% (4)	9.2% (15)		
Stage III	12.9% (8)	13.6% (129)	9.8% (72)	13.0% (7)	30.5% (50)		
Stage IV	6.5% (4)	6.0% (57)	3.0% (22)	3.7% (2)	20.1% (33)		
Unknown	0	0.1% (1)	0	0	0.6% (1)		

Grade 1	32.3% (20)	31.1% (296)	35.3% (259)	33.3% (18)	11.6% (19)	63.2% ^$^	63.6% ^$^
Grade 2	29.0% (18)	29.4% (280)	32.4% (238)	25.9% (14)	17.1% (28)	30.3% ^$^	29.5% ^$^
Grade 3	38.7% (24)	39.5% (376)	32.3% (237)	40.7% (22)	71.3% (117)	6.5 ^$^	6.9% ^$^

Endometrioid	74.2% (46)	70.8% (674)	77.4% (568)	66.7% (36)	42.7% (70)	76.9% *	
Serous	9.7% (6)	14.5% (138)	12.1% (89)	14.8% (8)	25.0% (41)	7.4% * Combined	
Clear Cell	4.8% (3)	2.2% (21)	1.9% (14)	1.9% (1)	3.7% (6)		
Adenosarcoma/Leiomyosarcoma	0	2.7% (26)	1.5% (11)	1.9% (1)	8.5% (14)	3.4% *	
MMMT	9.7% (6)	7.5% (71)	5.2% (38)	11.1% (6)	16.5% (27)	6.2% *	
Other	1.6% (1)	2.3% (22)	1.9% (14)	3.7% (2)	3.7% (6)		

Any Lymph node sampling/dissection	64.5% (40)	64.3% (612)	64.2% (471)	55.6% (30)	67.7% (111)		
Pelvic nodes	64.5% (40)	60.0% (571)	59.4% (436)	55.6% (30)	64.0% (105)		
Para-aortic	46.8% (29)	20.6% (196)	15.8% (116)	9.3% (5)	45.7% (75)		
Omentectomy/biopsy	17.7% (11)	23.0% (219)	14.4% (106)	24.1% (13)	61.0% (100)		

Median EBL (ml)	300	70	50	100	500		
Range (ml)	50–3800	0–8000	0–2500	10–8000	50–8000		
EBL Data unavailable (*n*)	2	2.6% (25)	3.0% (22)	3.7% (2)	0.6% (1)		

Median LOS	6	1	1	2	6		
Range LOS	1–59	0–84	0–84	0–17	1–42		
Return to Theatre <30 Days	1.6% (1)	0.7% (7)	0.5% (4)	1.9% (1)	1.2% (2)		
30 Day Mortality	0	0.6% (6)	0.1% (1)	1.9% (1)	2.4% (4)		

Required any blood Transfusion		5.4% (51)	1.8% (13)	13.0% (7)	18.9% (31)		
Required any ITU admission		15.5% (148)	7.2% (53)	18.5% (10)	51.8% (85)		

Conversion to open	21.4% (3/14)		0.5% (4)	24.1% (13)			
Post Operative 30 day Morbidity							
Clavien-Dindo Grade II		9.6% (91)	5.3% (39)	33.3% (18)	20.7% (34)		
Clavien-Dindo Grade III		1.5% (14)	1.0% (7)	5.6% (3)	2.4% (4)		
Clavien-Dindo Grade IV		0.4% (4)	0.4% (3)	0	0.6% (1)		
Clavien-Dindo Grade V		0.6% (6)	0.1% (1)	1.9% (1)	2.4% (4)		


**Table 2 cancers-14-05463-t002:** Robotic cohort—Surgical outcomes and BMI.

BMI	(*n*)	Median Age	Median EBL (Range)		Blood Transfusion	ITU Use	Median LOS Days (Range)		Conversion (*n*)	Reason for Conversion
<25	138	66	50 mL	(0–300)	0.72%	2.17%	1	(1–31)		
25–29.9	179	72	50 mL	(8–400)	1.12%	5.03%	1	(0–84)	1	Adhesions
30–34.9	151	69	50 mL	(10–800)	1.99%	1.32%	1	(0–11)	1	Adhesions
35–39.9	116	67	75 mL	(5–2500)	1.72%	6.90%	1	(0–17)	1	Vascular injury
40–44.9	67	64	75 mL	(10–940)	1.49%	11.94%	1	(0–5)		
45–49.9	44	65	80 mL	(15–1500)	4.55%	22.73%	1	(0–11)		
50+	27	60	150 mL	(20–1900)	7.41%	48.15%	1	(0–8)	1	Vascular injury

**Table 3 cancers-14-05463-t003:** Robotic Cohort—Association between Age and Median EBL/Length of stay (LOS).

Age	(*n*)	Median BMI	Median EBL (Range)		Blood Transfusion	ITU Use	Median LOS Days (Range)	
<55	88	33.41	50	(10–1900)	3.41%	5.68%	1	(0–8)
55–59	102	30.84	50	(0–400)	0%	8.82%	1	(0–9)
60–64	114	32.58	50	(10–1700)	0%	6.14%	1	(0–8)
65–69	117	33.43	50	(8–1300)	3.42%	6.84%	1	(0–31)
70–74	149	30.67	50	(10–2500)	1.34%	6.04%	1	(0–17)
75–79	84	28.28	50	(10–1500)	3.57%	10.71%	1	(0–11)
80+	80	29.03	50	(5–300)	1.25%	7.50%	1	(0–84)

**Table 4 cancers-14-05463-t004:** Comparison of Length of surgery between Robotic 2010–2019, Robotic2015–2019 (excluding any learning curve) other MIS and open surgery. ^a^ as some patients will have pelvic lymph node assessment only and others will also have para-aortic lymph node assessment/sentinel node assessment these numbers are not summative.

2010–2019	Overall	Robotic 2010–2019	Robotic 2015–2019	Other MIS	Open
Corpus Cancer primary surgery (*n*)	952	734 (77.10%)	487	54 (5.67%)	164 (17.23%)
Median Length of surgery (HH:MM)	02:53	02:54	02:37	02:54	02:50
Range	00:32–07:03	00:32–07:03	00:32–05:45	00:53–05:53	01:05–06:44
Data Missing (*n*)	66	22	19	13	31

Median Length of surgery no nodes (*n*)	02:37 (340)	02:41 (263)	02:27 (169)	02:15 (24)	02:31 (53)
Median length of surgery any nodes (*n*)	03:00 (612)	03:00 (471)	02:43 (318)	03:23 (30)	02:53 (111)
Median length of surgery with pelvic nodes not para-aortic nodes (*n*) ^a^	03:00 (387)	03:00 (326)	02:40 (205)	03:15 (25)	02:30 (36)
Median length of surgery with para-aortic nodes (*n*) ^a^	03:07 (196)	03:10 (116)	03:00 (85)	04:07 (5)	02:57 (75)

## Data Availability

The data presented in this study are available on request from the corresponding author.

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
