# Peer review of "Enhanced Recovery after Uterine Corpus Cancer Surgery: A 10 Year Retrospective Cohort Study of Robotic Surgery in an NHS Cancer Centre"

_cancers, 2022, doi:10.3390/cancers14215463_

Round 1

Reviewer 1 Report

Thanks for giving me the opportunity to review the manuscript by Uwins and co-authors. It is indeed a very impressive cohort of women treated with robotic-assisted laparoscopy during a decade at a single UK institution. Data derived from the departmental database confirm many other similar (but in most cases smaller) retrospective studies, analyzing the impact of robotic surgery on various surgical outcomes in gynecologic oncology. The authors acknowledge several important limitations but I have some serious concerns with the current study:

1.     I find it difficult to identify the rationale for conducting the study. There is no hypothesis nor clear rationale other than “to assess surgical and survival outcomes”. This sounds utterly explorative and in the light of the many previous studies (including meta-analyses and RCTs) providing similar data, I find the scientific rationale very vague. I am not sure which gap in knowledge this study aims to fill.

2.     The survival analyses are dubious as no proper control group has been presented. Unadjusted survival curves comparing 481 vs 38 patients provide very little information on the oncologic safety (if this was the purpose) of robotic-assisted surgery. Multiple factors will surely confound these analyses including time trend bias as most open procedures were performed in the early years of the cohort. In addition, the open group represents a completely different population than the robotic. The authors provide a rather lengthy comparison with outcomes from the LACE study, a discussion with low value for many reasons.

3.     Several potentially interesting surgical outcomes are missing including short- and long-term morbidity. Did the introduction of sentinel lymph node biopsy have any impact on outcomes (OT, complications, survival etc)?

4.     This database is surely a goldmine and I am certain that many interesting aspects of robotic-assisted surgery for endometrial cancer could be addressed. These could include learning-curve analyses, cost analyses or other underexplored areas. The current analyses simply do not provide any novel insights, despite the large number of cases.

Author Response

Comments and Suggestions for Authors

Thanks for giving me the opportunity to review the manuscript by Uwins and co-authors. It is indeed a very impressive cohort of women treated with robotic-assisted laparoscopy during a decade at a single UK institution. Data derived from the departmental database confirm many other similar (but in most cases smaller) retrospective studies, analyzing the impact of robotic surgery on various surgical outcomes in gynecologic oncology. The authors acknowledge several important limitations but I have some serious concerns with the current study:

  1. I find it difficult to identify the rationale for conducting the study. There is no hypothesis nor clear rationale other than “to assess surgical and survival outcomes”. This sounds utterly explorative and in the light of the many previous studies (including meta-analyses and RCTs) providing similar data, I find the scientific rationale very vague. I am not sure which gap in knowledge this study aims to fill.

We thank the reviewer for their comments. The aim of this study was to assess the surgical and survival outcomes for uterine corpus cancer since RSFT introduced robotic surgery as a central element of the ERAS pathway for gynaecological cancer surgery.The authors believe that this is an important piece of work as it gives valuable insight into real life data in a busy cancer centre where there has been significant transformation in clinical outcomes such as surgical morbidity (e.g. length of stay, blood loss, transfusion, use of intensive care and return to theatre). This has largely been achieved due to the greater uptake of minimally invasive surgery, in endometrial cancer, despite a largely unchanged cohort of patients.  The economic benefits of this transformation is not to be underestimated at a time when the NHS is at increased financial pressure.

  1. The survival analyses are dubious as no proper control group has been presented. Unadjusted survival curves comparing 481 vs 38 patients provide very little information on the oncologic safety (if this was the purpose) of robotic-assisted surgery. Multiple factors will surely confound these analyses including time trend bias as most open procedures were performed in the early years of the cohort. In addition, the open group represents a completely different population than the robotic. The authors provide a rather lengthy comparison with outcomes from the LACE study, a discussion with low value for many reasons.

We once again thank the reviewer for their comments.  This is not a comparative study between cohorts of patients who had open and different types of minimally invasive surgery, rather a descriptive account of surgical outcomes in the different patient cohorts.  This descriptive narrative has been made to emphasise the improved surgical morbidity outcomes following the introduction of the robotic program where there has been a marked transformation in the uptake of minimal access surgery for patients with uterine corpus cancer.  The survival analysis is presented to establish no significant adverse oncological impact following the introduction of the robotic program. We acknowledge that the comparison to LACE may not be beneficial as this is a retrospective study but has been included due to its well established bench mark in surgical endometrial cancer care. The most homogenous group to provide a comparison with the greatest scientific value by route of surgery is in those women with stage 1 endometrioid endometrial cancer. This is the data that has been presented.

  1. Several potentially interesting surgical outcomes are missing including short- and long-term morbidity. Did the introduction of sentinel lymph node biopsy have any impact on outcomes (OT, complications, survival etc)?#

We take this constructive suggestion on-board. We have moved the comparative surgical time from the supplemental data to the main manuscript. Departmental practice has developed over the last decade to now provide robotic surgical care for >90% of patients with endometrial cancer. As sentinel lymph node technique and principles have only recently become more uniform we hesitate for draw firm outcomes from this data. We aim to publish this in the future once morbidity and survival data is mature.

  1. This database is surely a goldmine and I am certain that many interesting aspects of robotic-assisted surgery for endometrial cancer could be addressed. These could include learning-curve analyses, cost analyses or other underexplored areas. The current analyses simply do not provide any novel insights, despite the large number of cases.

We thank the reviewer for commending us on our database and absolutely agree that subsequent papers will involve the various outcomes about robotic surgery that they have suggested. The economic cost following the introduction of the robotic program has been analysed at a trust level. The potential cost saving through cost modelling using the 2019-2020 national HES data and the departmental open, laparoscopic and robotic data has demonstrated a cost saving of £2442 when robotic surgery is compared to open and £651 when compared to laparoscopy. The reduction in post-operative 30 day complication rate by performing robotic procedures on these patients has also demonstrated a cost saving of £763 vs open and £161 vs laparoscopic procedures. Due to the reduction in conversion rates cost modelling has revealed a potential saving for £671 when comparing robotic vs laparoscopic procedures. These costs are based on our standard three arm robotic procedures performed for a woman with endometrial cancer using cadier forceps, bipolar and scissors, avoiding the use of a needle driver.  With the introduction of extended user pricing by intuitive surgical ltd since 2020 the cost of each procedure has reduce further by £219. We agree that this is valuable information and have added it to our discussion.

Reviewer 2 Report

The submitted manuscript analyzes a large retrospective cohort of robotic surgeries (n=734) performed for endometrial cancer in the UK. This is one of the strengths of the study. Another strength is the formal correctness of the main parts of the manuscript (methods, results). The shortcomings start with the authors' names: Why aren't the first names mentioned?

The first serious concern is the unbalanced comparison cohort, in which only 5.7% of patients underwent conventional laparoscopic surgery while 17% underwent open surgery. Also, I think the data would potentially allow for much more detailed results. For example, one of the most critical aspects to be analyzed are perioperative complications, which the submitted work reduces to just a few aspects: 30-day mortality, return to the operating room, need for a blood transfusion, stay in the intensive care unit, and conversion to open surgery. Despite a significant number of patients operated with robotic assistance, the study does not provide information on some relevant complications such as instrument-related, organ-related, stage-related complications (the classification systematically presented in the reference publication PMID: 34691301). From a practical point of view, it would be crucial to know whether the complications related to the first trocar placement, ureteral injuries or vaginal cuff dehiscences differed significantly in the analyzed patient group (RALS versus conventional MIS). If the data is available, it would be highly recommendableto include these relevant aspects in the analysis. If not, these shortcomings should be openly admitted and discussed in the manuscript. I propose moving Table 1 before the Figures and including the three important Tables S1-S3 and the interesting Figure S1 in the main manuscript.

Author Response

Comments and Suggestions for Authors

The submitted manuscript analyzes a large retrospective cohort of robotic surgeries (n=734) performed for endometrial cancer in the UK. This is one of the strengths of the study. Another strength is the formal correctness of the main parts of the manuscript (methods, results). The shortcomings start with the authors' names: Why aren't the first names mentioned?

Many thanks for your comments this has been amended.

The first serious concern is the unbalanced comparison cohort, in which only 5.7% of patients underwent conventional laparoscopic surgery while 17% underwent open surgery. Also, I think the data would potentially allow for much more detailed results. For example, one of the most critical aspects to be analyzed are perioperative complications, which the submitted work reduces to just a few aspects: 30-day mortality, return to the operating room, need for a blood transfusion, stay in the intensive care unit, and conversion to open surgery. Despite a significant number of patients operated with robotic assistance, the study does not provide information on some relevant complications such as instrument-related, organ-related, stage-related complications (the classification systematically presented in the reference publication PMID: 34691301). From a practical point of view, it would be crucial to know whether the complications related to the first trocar placement, ureteral injuries or vaginal cuff dehiscences differed significantly in the analyzed patient group (RALS versus conventional MIS). If the data is available, it would be highly recommendable to include these relevant aspects in the analysis. If not, these shortcomings should be openly admitted and discussed in the manuscript. I propose moving Table 1 before the Figures and including the three important Tables S1-S3 and the interesting Figure S1 in the main manuscript.

We thank the reviewer for taking the time to review our paper and their helpful suggestions.  Table 1 and the supplemental tables have been moved into the main manuscript. We have reviewed our data and have found no significant incidence of instrument related injury, visceral injury or vaginal cuff dehiscence to establish any significant trends. We have added 30day morbidity data using the Clavien-Dindo classification to table 1.

The manuscript is intended as a descriptive narration of patient outcomes over the last decade who have been treated for endometrial cancer by different modalities of surgery. There is no superiority or inferiority comparison intended between the modalities as the uptake of each approach has significantly changed with the embedding of the robotic program and hence this would make this an in-balanced comparison as suggested by the reviewer. We have made changes to the manuscript to clarify this.

Reviewer 3 Report

This is a well-written paper assessing the surgical and survival outcomes for robotic surgery uterine corpus cancer in a single center. I would make the following comments regarding the paper:                                   

1.      Please show the operation-related complication of both robotic and open cohorts by using Clavien-Dindo classification.

2.      Please analyze survival outcomes in patients with high-risk uterine cancer, including advanced stage, high grade, and high-risk histologies.

Author Response

This is a well-written paper assessing the surgical and survival outcomes for robotic surgery uterine corpus cancer in a single center. I would make the following comments regarding the paper:                                   

  1. Please show the operation-related complication of both robotic and open cohorts by using Clavien-Dindo classification.

Many thanks for your suggestion.  The presented outcomes have been classified by the Clavien-Dindo classification as suggested and added to Table 1

  1. Please analyze survival outcomes in patients with high-risk uterine cancer, including advanced stage, high grade, and high-risk histologies.

We have now presented data as requested for the robotic cohort regardless of tumour type or grade by stage. This is presented for the robotic group only as the numbers in the other cohorts are too small to draw any meaning full conclusions.

Round 2

Reviewer 1 Report

The criticism I raised after reviewing the first version of the manuscript remains after reading the revised version:

1.     The introduction should ideally give the reader a good understanding why the study is needed. The structured background underscores the well-known merits of robotic surgery for endometrial cancer (documented in numerous papers including several meta-analyses/systematic reviews): shorter length of stay, less conversions, less blood loss, ability to perform staging in frail and obese. However, the aim of the study was “to assess surgical and survival outcomes for uterine corpus cancer”. This is clearly not novel and adds very little to current knowledge (but important for internal quality assessment).

2.     Uncontrolled survival analyses have limited or no value. 

3.     Unadjusted comparisons (rob/MIS vs open) have limited or no value.

4.     The change of title is very confusing. ERAS is a concept with multiple components outlined by international guidelines. MIS is definitely a key component but MIS alone does not equal ERAS. If MIS was the only component in the ERAS-program (I could not find any other), the title is misleading and devaluates analyses of properly introduced ERAS-programs.

Author Response

Thank you for taking the time to review our paper. I am pleased with the final result and grateful for the time you have spent to enable it to be as good as it can be. In response to your final points:

  1. The aim of this study was to assess the surgical and survival outcomes for uterine corpus cancer since Royal Surrey NHS Foundation Trust introduced robotic surgery as a central element of the ERAS pathway for gynaecological cancer surgery. A retrospective cohort study was therefore performed of patients undergoing surgery for uterine corpus cancer between the 1st January 2010 and the 31st December 2019 to evaluate the success of the robotic program. This paper presents the experience of implementing robotic surgery for uterine corpus cancer in a high volume UK cancer centre with detailed breakdown of both stage and grade of disease and “real world” survival statistics.
  2. This paper presents all sequential surgical treatment for primary corpus cancer between 2010 and 2019. The 3 defined cohorts, robotic surgery, open Surgery and other MIS define the type of surgery performed via this route in this time period. They are not comparison groups. We have considered your point.  We could: 1) remove all the survival curves from the paper but in our opinion this will reduce the quality of the paper, 2) compare our survival with a historical cohort who have had open surgery or that of the concurrent cohort who have had open surgery. We considered this but for a historical cohort our series would be small and be subject to bias and the comparison be of limited or no use.  Comparison with the open and other MIS group would not be valid as those who had open surgery, particularly in the early years of the study, will have had higher risk disease or more complex surgery and had a poorer prognosis as a result.  We have therefore chosen to present our five year overall and relapse-free survival, stratified by stage for the robotic cohort.  Direct comparison to published studies is imperfect, partly due to not being able to compare small subgroups with available population survival data but these internationally recognised studies do provide us with a benchmark for us to compare our local outcomes and is presented in the discussion 
  3. This paper presents all sequential surgical treatment for primary corpus cancer between 2010 and 2019. The 3 defined cohorts, robotic surgery, open surgery and other MIS define the type of surgery performed via this route in this time period. They are not comparison groups.

  1. Minimally invasive surgery is one of the key principals of enhanced recovery after surgery (ERAS). Royal Surrey NHS Foundation Trust was an early adopter and pioneer of ERAS. The ERAS program and associated pathways were established by the time the first robotic system was installed.  Robotic surgery was implemented to further support increased access to minimally invasive surgery as a central element of the ERAS pathway for gynaecological cancer surgery.

Reviewer 2 Report

The authors performed a careful revision. The structure of the manuscript benefited from the changes. Some corrections apparently could not be made due to limited data. I have no further comments.

Author Response

Thank you so much for taking the time to review our paper.  I am really pleased with the final result and grateful for the time you have spent to enable it to be as good as it can be.  

Reviewer 3 Report

Thank you for proper revision.

Author Response

(The authors gave the same response as above.)
